# Implicit Neural Representations with Levels-of-Experts

**Zekun Hao**\*†

hz472@cornell.edu

**Arun Mallya**\*
\*NVIDIA

amallya@nvidia.com

**Serge Belongie**†
†Cornell University
sjb344@cornell.edu

**Ming-Yu Liu**\*

mingyul@nvidia.com

## Abstract

Coordinate-based networks, usually in the forms of MLPs, have been successfully applied to the task of predicting high-frequency but low-dimensional signals using coordinate inputs. To scale them to model large-scale signals, previous works resort to hybrid representations, combining a coordinate-based network with a grid-based representation, such as sparse voxels. However, such approaches lack a compact global latent representation in its grid, making it difficult to model a distribution of signals, which is important for generalization tasks. To address the limitation, we propose the Levels-of-Experts (LoE) framework, which is a novel coordinate-based representation consisting of an MLP with periodic, position-dependent weights arranged hierarchically. For each linear layer of the MLP, multiple candidate values of its weight matrix are tiled and replicated across the input space, with different layers replicating at different frequencies. Based on the input, only one of the weight matrices is chosen for each layer. This greatly increases the model capacity without incurring extra computation or compromising generalization capability. We show that the new representation is an efficient and competitive drop-in replacement for a wide range of tasks, including signal fitting, novel view synthesis, and generative modeling.

## 1 Introduction

There has been a growing interest in representing low-dimensional but high-frequency signals, such as images, videos, and 3D scenes, with fully-connected neural networks. A common paradigm is to use a coordinate-based multilayer perceptron (MLP) that takes coordinate positions as input and predicts the data value at the specified location [30, 44, 49]. Compared to explicit representations such as point clouds and voxel grids, this kind of implicit neural representation (INR) is memory efficient and can model a distribution of signals for conditional synthesis tasks [2, 26, 34, 37, 43] thanks to its ability to learn a compact and meaningful latent space.

However, scaling up an INR to better represent higher-resolution signals or a distribution of signals, like a distribution of images, is challenging because the mapping can be highly nonlinear. To increase the model capacity to deal with the complexity, we can either make the MLP wider by increasing the dimensions of activations or deeper by stacking more layers. Unfortunately, both options will dramatically increase the computation needed at each data point. Recently, Rebain *et al.* [40] have shown that this results in an undesirable trade-off because the representation power gain diminishes quickly with increased width or depth. We further explore and analyze this issue in Section 4.2.

Many recent works bypass this scaling problem by using a hybrid representation [4, 8, 10, 11, 19, 24, 25, 33, 38, 40, 41, 47, 48]. A discrete data structure, such as sparse voxels, decomposes the space into grids. Within each grid, a lightweight MLP conditioned on the grid embedding produces local detail at a scale finer than the grid resolution. However, such an approach has two major limitations: (1) The reliance on smooth interpolation of grid embeddings [24, 25, 33, 38, 47] or

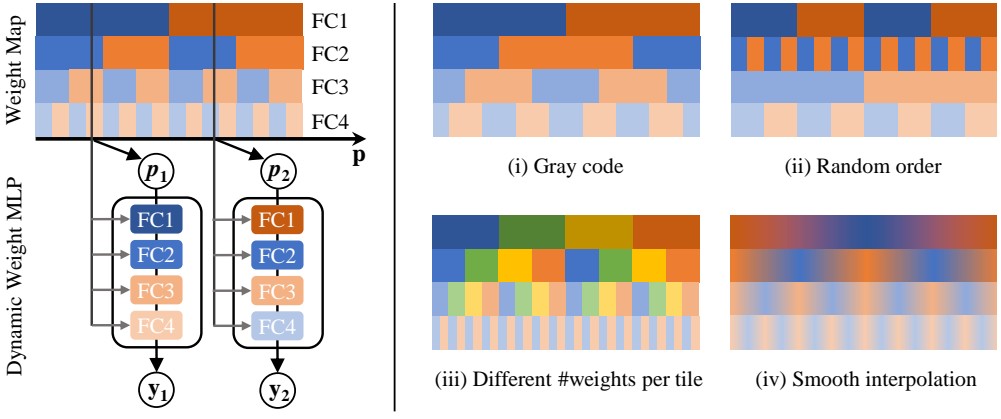

(a) MLP with position-dependent weights       (b) Examples of different weight tiling patterns

Figure 1: The Levels-of-Experts (LoE) framework. (a) A position-dependent MLP with 1D input, $y = f(x, \theta(x))$ (activation functions omitted for brevity). In this example, each fully-connected (FC) layer has two candidate weight matrices (marked in blue and orange, shade denotes layer depth), arranged in a periodical and hierarchical manner. According to the input location, one of the weight copies is selected for each layer. (b) A variety of hierarchical tiling patterns can be used with LoE. They are not confined to (i) a specific alignment, and can have varying (ii) orders of granularity, or (iii) length of the repetend, and (iv) can be generalized to a smooth interpolation across the weight matrices. Thus, each layer at a different *level*, or scale, has a number of *experts* with their own weight matrices, specializing at different regions of the input space.

output-domain tiling and blending [4, 19, 28, 48] to encourage continuity across the grid boundaries can negatively impact computation efficiency; and, (2) As the underlying signal is described by multiple distributed features stored in a grid, which are associated with fixed locations, it lacks a global, compact latent representation unless it employs another expensive model to generate the grid embedding itself [5, 38, 42]. Still, this is only possible in limited cases, *e.g.*, regular grid, without any sparsity, pruning, or hashing.

Our approach extends the idea of hybrid representation by storing the weights of an MLP on a multi-resolution tiled grid. Conceptually, for each of the linear layers, we assign multiple independent copies of its weights, arranged in a tiling pattern and repeated to fill the space as visualized in Figure 1. This partitions the space so that each copy of weights only needs to handle inputs within certain periodical intervals, essentially making the weights of the MLP position-dependent. We refer to this proposed framework as Levels-of-Experts (LoE). Each layer at a different depth, *level*, has a number of *experts* with their own weight matrices, specializing in different partitions of the input space, depending on the tiling pattern used.

Similar to Fourier features [30, 49], we use different grid resolutions for each layer or depth, so that the weight of different layers repeats at different frequencies. This arrangement has several desired properties: (i) While the weight of each layer is repeating, the learnable combined parameterization of all the layers helps avoid repetition over the input range of interest, and (ii) A large number of uniquely parameterized intervals of the combined model can be obtained and tailored to the underlying problem. In fact, in Section 4.1, we show that our model can fit a scene even without any input position encoding – the position-dependent weight can itself serve as a form of positional encoding! We show that when compared to dividing the space to use different MLPs [40, 41], our layer-level tiling approach can reduce output discontinuities without relying on computationally expensive smooth interpolation or blending, while at the same time improving the representation power. Finally, compared to non-repeating grid embeddings [24, 25, 47], our approach encourages the learning of generalizeable mapping, as shown in Sections 4.1 and 4.2, and also improves parameter efficiency, particularly benefiting generative modeling tasks, as shown in Section 4.4.

Our model has a computational cost comparable to a regular MLP of identical architecture, at the same time being more expressive. Although the parameter count, and thus representation power, is greatly amplified by the use of position-dependent weights, only a single copy of weights is active at each input. In practice, this can be implemented efficiently with an off-the-shelf fused gather-GEMM-

scatter operator [1] with little speed loss. Our method is a drop-in replacement in many applications without the need for any further modifications. To summarize, we make the following contributions:

1. We introduce a novel hybrid implicit neural representation that is parameterized by a hierarchy of position-dependent and periodic weights (Section 3).

2. We extensively study the effect of various design decisions including the periodicity and hierarchy of weights, weight interpolation methods, and the use of input encodings (Section 4).

3. We demonstrate the efficiency and representation power of our architecture on challenging tasks including high-resolution image fitting, video fitting, novel-view synthesis, and image generation (Section 4).

## 2   Related Work

**Implicit neural representations (INRs).** INRs represent a signal with a pointwise (coordinate-based) neural network that takes a coordinate as the input and predicts the data value at the location specified by the input coordinate. With INRs, one can query continuous locations independently and efficiently, which is a desired property for many learning, graphics, and vision tasks [50]. INRs have been used in representing images [2, 44, 46, 49], shapes [13, 29, 37, 44, 49], and scenes [3, 30, 35, 36, 45]. Earlier INRs were based on MLPs with ReLU activations [37, 46] and often failed to represent high-frequency detail in the underlying signal. Recent works have greatly addressed the issue by leveraging better input encoding designs [30, 49], activation functions [39, 44], or network architectures [9, 23]. Our approach, a position dependent MLP architecture, is orthogonal to these approaches and can potentially be used in conjunction with them to achieve better results.

**Hybrid representations.** Several works propose combining INRs with explicit discrete structure representations to improve both the computation and memory efficiency for modeling large and complex signals. Such a hybrid approach often partitions the input space into smaller regions based on the adopted discrete structure representation, which results in local parameterization, or decomposes the input space to low-rank subspaces [5, 6]. Various discrete structure representations for local parameterization have been explored, including regular grids [4, 19, 33, 38, 41], sparse voxels [15, 24], voronoi cells [40], octrees [25, 47], convex parts [8], and learned shape elements [10, 11]. We can even use another neural network to predict the parameterization, which enables generalization [5, 38, 42]. With the hybrid approach, one first obtains a local feature with the discrete structure and the coordinate, which is then inputted to the pointwise MLP to get the data value.

Note that the discrete nature of the hybrid representation calls for special and often costly designs to ensure smooth transitioning across the subdivision boundaries. One popular approach is to smoothly interpolate the grid feature [24, 25, 33, 38, 47] before handing it off to the MLP. Such an approach only requires one MLP evaluation per sample, but the cost of interpolation can be high for high-dimensional features on a high-dimensional grid. Several approaches [4, 19, 28, 48] allow the MLP to predict signals beyond the grid boundaries so that multiple predictions of the same coordinate from nearby grids can be evaluated and smoothly blended in the output domain, but they are more expensive to compute. There are also hybrid representations that completely abandoned smooth interpolation and achieved considerable speedup [41]. However, it requires distillation from a larger, pretrained network to mitigate discontinuity artifacts. Our approach is also a hybrid approach. It enjoys computation and memory efficiency but does not suffer from the boundary interpolation issue.

**Neural networks with input-dependent weights.** Our method can be regarded as a special type of hybrid representation that use a multi-level tiled grid to parameterize the weights of the pointwise MLP. Depending on where the input coordinate lies on each level of the grid, a different combination of weights is used for the network. Reiser *et al.* [41] shares the same high-level idea of having coordinate-dependent network weights, but they learn completely disjoint networks for different grid locations, while we use a hierarchical and periodical structure. We will show the importance of our hierarchical and periodical structure in obtaining a smooth and expressive representation.

In a broader context, neural networks with data-dependent weights have been used for modeling 3D animation [7, 17] as a form of a collection of experts and for solving differential equations [32] to represent solutions. Our work is different as we use a layer-wise hierarchical parameterization and is designed for hybrid neural implicit representation.

## 3 Method

A typical coordinate-based multi-layer perceptron (MLP) can be described as a stack of layers,

$$\hat{f} : \mathbf{p} \to (g^k \circ \phi \circ g^{k-1} \circ \cdots \circ \phi \circ g^1 \circ \gamma)(\mathbf{p}), \tag{1}$$

where $\mathbf{p}$ is the input coordinate at which the MLP is being evaluated, $\gamma$ is an input mapping, such as the sine-cosine positional encoding [30], $\phi$ is a non-linear activation function, and $g^i : \mathbf{x} \to \mathbf{W}^i\mathbf{x} + \mathbf{b}^i$ is the $i^{th}$ linear layer, which performs an affine transformation on the input $\mathbf{x}$, parameterized by a weight matrix $\mathbf{W}^i$ and a bias vector $\mathbf{b}^i$. During training, $\mathbf{W}^i$ and $\mathbf{b}^i$ are optimized via gradient descent to fit the MLP to the data.

In our Levels-of-Experts (LoE) approach, instead of regarding each $\mathbf{W}^i$ as a single learnable matrix, we additionally model it as a function $\psi^i(\cdot)$ of the input coordinate $\mathbf{p}$. The resulting dynamic-weight linear layer has the form, $h^i : (\mathbf{x}, \mathbf{p}) \to \psi^i(\mathbf{p})\mathbf{x} + \mathbf{b}^i$, where $\mathbf{x}$ are the inputs to the layer, and $\mathbf{p}$ are the location at which the MLP is being evaluated. By replacing the traditional linear layers $g^i$ in the MLP with dynamic-weight layers $h^i$, we obtain an MLP with input-dependent weights,

$$f : \mathbf{p} \to (h^k(\mathbf{p}) \circ \phi \circ h^{k-1}(\mathbf{p}) \circ \cdots \circ \phi \circ h^1(\mathbf{p}) \circ \gamma)(\mathbf{p}). \tag{2}$$

As the resulting position-dependent weight matrix has a much higher dimension compared to its input and output vectors and will be evaluated at a large number of query points, it is important for the weight generation functions $\psi^i(\mathbf{p})$ to be fast, inexpensive, and yet expressive. This rules out the popular weight-prediction networks used in hypernetwork-based approaches [14], in which one has to predict a high-dimensional weight per position. Instead, we use a simple, lightweight function, specifically a coordinate interpolation-based method. Multiple candidate values for the weight matrix are stored in a regular grid (tile) and interpolated in a cyclic manner based on the input coordinates.

Consider the case where we have a grid containing $N$ matrices $\{\mathbf{W}^i_0, \ldots, \mathbf{W}^i_{N-1}\}$, where $i$ is the layer depth, and $N$ is a nonnegative integer. We are only interested in the case that $N > 1$ as $N = 1$ reduces to the original pointwise MLP formulation. Given a 1D coordinate $\mathbf{p} = (p)$, the input-dependent weight for layer $i$, $\mathbf{W}^i$, is computed as

$$\mathbf{W}^i = \psi^i(\mathbf{p}) = \psi^i(p) = \sum_{j=0}^{n-1} B_{j,N}(\alpha^i p + \beta^i)\mathbf{W}^i_j. \tag{3}$$

where $\alpha^i$ and $\beta^i$ are hyperparameters that adjust the scale and translation of the grid for each layer and $B_{j,N}$ is the blending function that computes the blending coefficient for the $j$-th candidate. The blending coefficient can take many different forms. For linear and nearest interpolations, they are defined as follows:

$$B_{j,N}^{\text{linear}}(q) = \max(0, 1 - |(q + 1 - j) \bmod N - 1|) \tag{4}$$

$$B_{j,N}^{\text{nearest}}(q) = \begin{cases} 1 & \lfloor q \rfloor \bmod N = j \\ 0 & \text{otherwise.} \end{cases} \tag{5}$$

Note that here $\bmod$ denotes positive remainder operation: $a \bmod b = a - b\lfloor \frac{a}{b} \rfloor$. We also note that the above equations can easily be extended to multi-dimensional coordinate spaces.

For linear interpolation, regardless of the tile resolution, only 2 of the blending coefficients are non-zero for each coordinate in our 1D example. On the other hand, the nearest interpolation scheme only has a single non-zero coefficient for each coordinate. This sparsity allows a fast and efficient implementation of performing the dynamic-weight linear layer computation for batched inputs: for each candidate weight matrix, $\mathbf{W}^i_j$ where we only gather input vectors that have $B_{j,N} > 0$ at a time, perform matrix multiplication and scaling, and finally scatter the results to the output matrix.

Empirically, we find it helpful to have different layers of our MLP with different spatial frequencies on the grid. This can be easily achieved by using a different set of $\alpha^i$ and $\beta^i$ per layer. Using different frequencies at different layers gives an inductive bias to the MLP to capture different repetition patterns. It also serves as a form of regularization that encourages the learning of smooth mapping via weight sharing at different locations. We show this is particularly useful in reducing artifacts for novel view synthesis tasks (see Section 4.3).

A non-exhaustive list of potential grid arrangements—a grid arrangement corresponds to a set of $\{(\alpha^i, \beta^i)\}$—are presented in Fig. 1. We note it is even possible to use a randomized tiling pattern by

transforming the grid with a random affine transformation, while still seeing a significant performance gain compared to a regular MLP, as evident by the *Random Affine* experiment in Table 1. Unless otherwise mentioned, we arrange the grids in a progressively growing fashion throughout the paper. We start with the first grid (corresponding to the first MLP layer) covering the full input space without repetition, and progressively subdivide the grids using additional layers. This is shown in Fig. 1(a). This arrangement partitions the input space into uniform-sized grids, with each one having a unique combination of weight matrices. A comparison of different grid arrangements is included in the supplementary material.

## 4   Experiments

In this section, we validate LoE on 4 challenging tasks. In the first two experiments, we fit our model to high-resolution image and video data, evaluate its performance, and study the effect of various design components. Then in Section. 4.3, we evaluate our model on the indirectly supervised, novel-view synthesis task and study its inductive bias. Finally, in Section. 4.4, we demonstrate its generalization capability by training a generative adversarial network (GAN). All the code will be made publicly available.

### 4.1   Fitting to a High-resolution Image

We study the effect of our hierarchical weight tiling on model capacity and computational efficiency by fitting networks to a high-resolution image of size $8192 \times 8192$ [25] pixels. An image is considered as a set of pixels $\{(\mathbf{p}_i, \Theta(\mathbf{p}_i))\}$ represented by their 2-D coordinates $\mathbf{p}_i = (x_i, y_i)$ and RGB colors $\Theta(\mathbf{p}_i) \in \mathbb{R}^3$. The model $\mathbf{p} \to f(\mathbf{p})$ takes the coordinate as input and predicts the color at the given coordinate. The goal is to fit the model to the data by minimizing the loss: $\mathcal{L}_2 = \sum_i \|f(\mathbf{p}_i) - \Theta(\mathbf{p}_i)\|^2$.

Table 1 compares our model with several baseline methods and ablations. Our main method significantly outperforms baseline methods that do not use position-dependent weights. Despite sharing the same network architecture and computational cost, an MLP using the sine-cosine positional encoding (PE) as the input mapping [30] performed 12dB worse than our model. We also compare with a hybrid model that learns an input coordinate embedding (CE) [2] for the MLP. Their fitting quality is significantly lower than ours at the same parameter count while incurring a higher computational cost. This suggests that learning a position-dependent weight is more effective than learning a grid of embeddings.

We believe that the effectiveness of our method partially comes from the use of a hierarchical and periodic tiling pattern, which encourages the learning of periodic, spatially-shared features. In Table 1, the *interleaved* model uses a periodic weight tiling scheme but lacks the multi-scale arrangement (Fig. 2(d)). Effectively, multiple independent MLPs are learned, each handling a pixel-skipped subset of the image. This only improves the performance slightly compared to the PE MLP baseline. On the other extreme, in the *chunked* experiment, we partition the input space into uniformly sized chunks and use independent MLPs to handle each chunk. Although the fitting quality is improved, there are large variations in errors across different chunks, as shown in Fig. 2(c). In fact, the chunk boundaries are visible in the fitted image, indicating continuity issues.

Our method achieves the best performance by having the tiled weights repeat at a wide range of intervals. This allows a more efficient data representation by exploiting periodicity at a wide range of frequencies and allows a more adaptive distribution of model capacity and fitting error that is less dependent on the geometry of the data. We also compare our piecewise constant weight parameterization against the smooth, piecewise linear variant, implemented by the bilinear interpolation of weights, shown in Fig. 2(b). Despite having 4 times the computational cost, the fitting quality of the smooth variant is only slightly better (+0.61dB) than the faster, piecewise constant version while also sharing a similarly homogeneous error distribution. This indicates that by using the piecewise constant parameterization, the full performance of the tiled weight models can be enjoyed at only a fraction of the cost. Surprisingly, the tiled weight model is able to achieve reasonable performance even without the use of any input position encoding. In the *Constant Input (no PE)* experiment shown in Table 1, we feed a constant vector to the first layer instead of the coordinate encoding. In fact, the position-dependent tiled weight itself is already a form of positional encoding. It is able to identify a large number of unique intervals in the input space (up to $\prod_i n_i$, where $n_i$ is the number of candidate

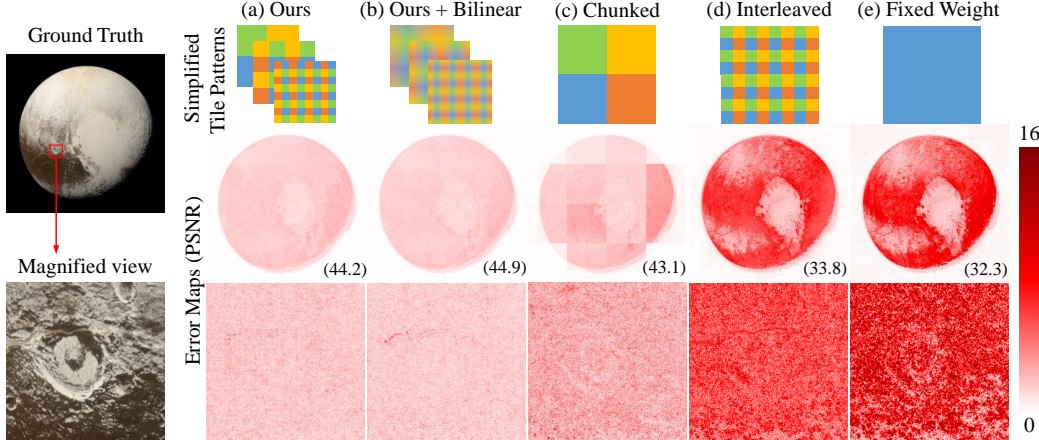

Figure 2: Comparison of errors while fitting to a 64MP image. Our method with discontinuous weights (a) has low and uniformly distributed error comparable to (b), the more expensive version that bilinearly interpolates the weights. (c) and (d) use an ensemble of networks without hierarchical weight tiling, resulting in high error variation, discontinuities, and low fitting quality.

Table 1: Comparison of parameter count, computational cost in number of multiply-accumulates (MACs) per sample, and fitting quality on the 64MP color image of Pluto shown above [25]. All the models use the same number of layers and hidden channels. The model size of our method is chosen to be comparable to ACORN [25]. For the coordinate embedding (CE) baseline, we evaluate multiple grid resolutions and report the best result.

| Shown in Fig. 2 | Periodic Tiling | Multi-scale | | Parameters | MACs | PSNR (dB) | SSIM |
|---|---|---|---|---|---|---|---|
| | | | PE MLP [30] | 0.59M | 0.57M | 32.34 | 0.869 |
| | | | PE + CE [2] | 9.37M | 0.65M | 39.65 | 0.967 |
| (d) | ✓ | ✗ | Interleaved | 9.37M | 0.57M | 33.80 | 0.876 |
| (c) | ✗ | ✗ | Chunked | 9.37M | 0.57M | 43.13 | 0.980 |
| | ✓ | ✓ | Random Affine | 9.37M | 0.57M | 42.08 | 0.973 |
| | ✓ | ✓ | Constant Input (no PE) | 8.92M | 0.56M | 39.48 | 0.955 |
| (b) | ✓ | ✓ | Ours Bilinear | 9.37M | 2.28M | **44.85** | **0.985** |
| (a) | ✓ | ✓ | Ours | 9.37M | **0.57M** | **44.24** | **0.983** |

weights of layer $i$). Related ideas of using periodical weights in a coordinate-based network are also found in MFN [9] and BACON [23].

## 4.2 Fitting to a Video

We fit our model to a video [44] with 300 frames and a resolution of $512 \times 512$. In this case, each pixel in the video is associated with a 3D coordinate $\mathbf{p}_i = (x_i, y_i, t_i)$. The quantitative results are shown in Table 2, and visual comparisons in Fig. 3. Compared to fixed-weight models such as PE MLP or SIREN [44], the capacity of our model grows favorably with increased input dimensions, without incurring extra computation. For higher-dimensional problems, we can simply use higher dimensional weight tiles. In this case, we use a combination of $2^3$ and $4^3$ tiles, which amplifies the number of parameters by $8\times$ and $64\times$ compared to a regular linear layer with the same number of channels. This cannot be done in a fully implicit model without greatly increasing the computation. For example, in the SIREN-L experiment, we attempt to increase the model capacity of SIREN by quadrupling the hidden channel count. The resulting model needs $15\times$ more computation, yet the quality is still lacking. This confirms the diminishing return phenomenon associated with coordinate MLPs [40]. Compared to embedding-based hybrid representations such as coordinate embedding (CE), which in this case include a dense $64^3 \times 64$ grid that store the position-dependent features and

trilinearly interpolated, the conclusion in Sec. 4.1 still holds that our approach performs significantly better under the same parameter count.

Table 2: Comparison of model size, computation and fitting quality on a short video. All of the models have 4 hidden layers and 256 hidden channels, except for SIREN-L, which has 1024 hidden channels.

|         | Params. | MACs  | PSNR(dB) |
|---------|---------|-------|----------|
| PE [30] | 279K    | 0.28M | 27.33    |
| PE + CE [2] | 17.1M | 0.29M | 35.83  |
| SIREN [44] | 265K | 0.26M | 29.13    |
| SIREN-L | 4.21M   | 4.20M | 37.71    |
| Ours    | 16.9M   | 0.28M | **39.98** |

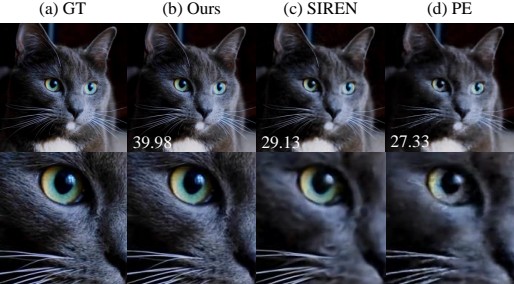

Figure 3: Visual comparison of video fitting results. Numbers indicates PSNR (dB). All the models have the same computational cost.

## 4.3 Novel View Synthesis

So far our LoE model has demonstrated improved quality in fitting to high-resolution signals via direct supervision. Here, we examine if the model has the desirable inductive bias to work in an under-constrained setting with indirect supervision. We evaluate our method on a novel view synthesis task, where the network models color $\mathbf{c}$ and opacity $\sigma$ at each 3D location and under different view directions. This kind of volumetric 3D representation is also known as neural radiance fields (NeRF) [30]. Given an image, each pixel in the image can be associated with a ray $\mathbf{r}(t) = \mathbf{o} + t\mathbf{d}$. The color of the pixel can be obtained by sampling points $t_i$ along the ray, querying the neural network at these points to obtain color and opacity $\mathbf{c}_i, \sigma_i = f(\mathbf{r}(t_i), \mathbf{d})$, and perform volumetric rendering via numerical integration [27]:

$$\mathbf{C}(\mathbf{r}) = \sum_{i=1}^{N} T_i(1 - \exp(-\sigma_i \delta_i))\mathbf{c}_i, \text{ where } T_i = \exp\left(-\sum_{j=1}^{i-1} \sigma_j \delta_j\right), \text{ and } \delta_i = t_{i+1} - t_i. \quad (6)$$

The training is done by minimizing the photometric loss between the rendered colors and ground truth pixel values: $\mathcal{L}_2 = \sum_k \|\mathbf{C}(\mathbf{r}_k) - \mathbf{C}_k\|^2$.

We compare our method with baselines on the Tanks and Temples dataset [21, 24], which contains 133 training images at a resolution of $1920 \times 1080$. All the models compared have 4 hidden layers and 256 hidden channels. We present the quantitative results in Table 3 and include a visual comparison in Fig. 4. As shown in the zoom-in view, the use of hierarchical, position-dependent weights significantly sets the otherwise identical PE MLP baseline apart by reproducing much better detail.

It has been observed that when the input space is partitioned into grids, and independent MLPs are learned for each grid location, there will exhibit significant free space artifacts in the result [41]. Our model does not have such a problem, despite similarly having position-dependent weights. To gain a better understanding, we experimented with a model that lacked hierarchical arrangement in its weight grids. Despite having a lot more parameters, the ablated model, which indeed suffering from free space artifacts, performed far worst than the baseline PE MLP (Fig. 5). This shows that the use of hierarchically tiled weights is important for providing a good inductive bias for the task.

## 4.4 Image Generation with GANs

In this section, we demonstrate the generalization capability of the LoE model on the challenging image generation task. The coordinate-based models are used as the generators for the generative adversarial networks (GAN) [12]. More specifically, the model $f(\mathbf{p}, \mathbf{z})$ takes both a coordinate $\mathbf{p}$ and a noise vector $\mathbf{z}$ as input, and map them to a color value. To generate an image, a fixed $\mathbf{z}$ is used and the network is queried at every pixel location $\mathbf{p} \in \mathbf{P}$. We denote the process of generating a full image as $G(\mathbf{z}) = \{f(\mathbf{p}, \mathbf{z}) | \mathbf{p} \in \mathbf{P}\}$. Images of different appearances can be generated by sampling the noise vector from a fixed distribution $\mathbf{z} \sim p_{\mathbf{z}}$. An additional discriminator network $D$ is used to provide the training signal. We use hinge loss [22] as the GAN objective.

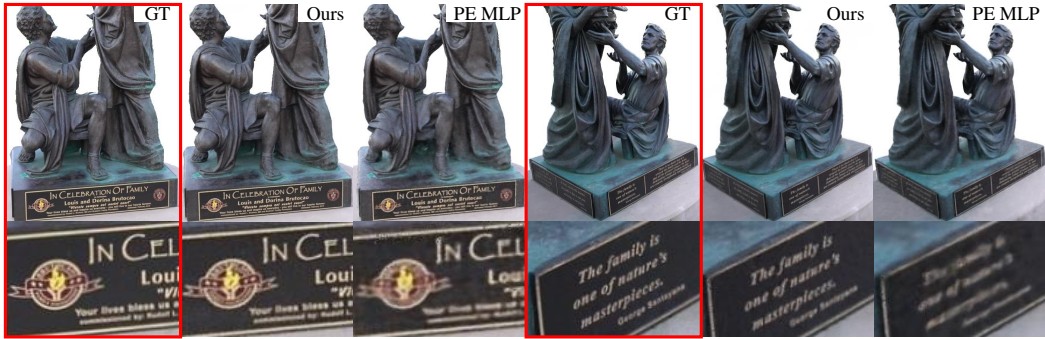

Figure 4: Visual comparison of novel view synthesis results from our model as well as the baseline model. Below each image there are local crops that better show the detail. Our model produces extremely sharp detail compared to the baseline while having the same computational cost. Both models share the same model architecture, with the only difference being the use of hierarchical weight tiling.

Table 3: Comparison of novel view synthesis quality on the *Family* scene. All the models have the same computational cost of 315KMACs per sample.

|  | #Params. | PSNR(dB)↑ | SSIM↑ |
|---|---|---|---|
| PE MLP [30] | 317K | 30.50 | 0.900 |
| No Hier. | 17.8M | 27.73 | 0.861 |
| Ours | 17.8M | **31.46** | **0.936** |

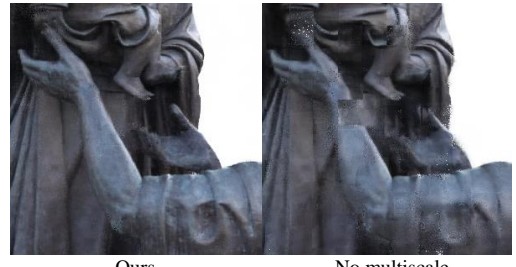

Ours        No multiscale

Figure 5: Free-space artifacts in the ablated model that uses tiled weights but lacks the multiscale arrangement. The weight grids in all the layers are aligned and repeated at the same interval.

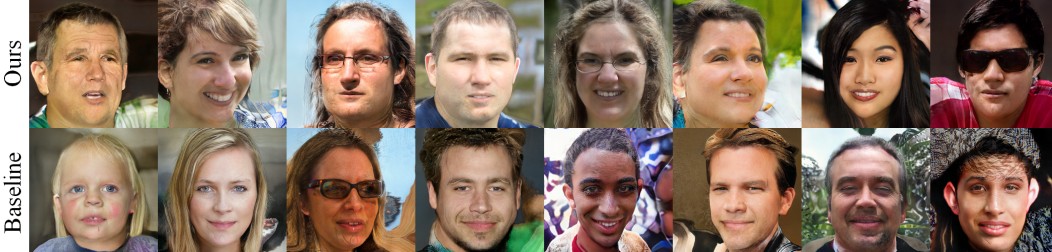

Figure 6: Comparison of image generation results on FFHQ dataset. Images in the top row are generated by our model. Images in the bottom row are generated by a baseline model with coordinate embedding. Both models have comparable parameter counts and computational costs.

To reduce computation and encourage easy reproduction, we use a simplified setting with lightweight components. For the generator, we use an 8-layer network with residual connections. The noise vector is directly fed into the first layer. For the discriminator, we use a multi-resolution patch discriminator [18] with spectral normalization [31]. The model is trained on the Flickr Faces-HQ (FFHQ) dataset [20] at a resolution of $256 \times 256$. For the baseline method, we use a coordinate embedding of $256 \times 256$ resolution and 256 channels in order to obtain a parameter count comparable to the LoE model. Please refer to the supplementary material for full implementation detail and larger-scale experiments.

We report the model size, computational cost and image quality measured in Fréchet inception distance (FID) [16] in Table 4, and provide a gallery of sample images in Fig. 6. Our method not only achieves a better FID but also produces images free of fixed noise pattern artifacts (shown in Fig. 7).

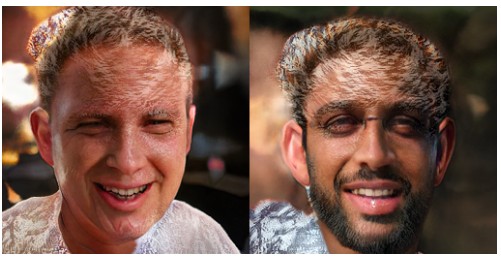

Figure 7: Examples of the fixed noise pattern artifacts in the images generated with the baseline PE + CE method. Our method does not suffer from this issue.

Table 4: Comparison of image generation quality on FFHQ dataset.

|            | #Params | MACs  | FID  |
|------------|---------|-------|------|
| PE + CE [2] | 19.9M   | 1.39M | 23.5 |
| Ours       | 19.6M   | 1.26M | **18.3** |

## 5    Discussion

In this work, we demonstrated a new type of hybrid implicit representation, called Levels-of-Experts (LoE), which is parameterized by hierarchical and periodic, position-dependent weights that are arranged on levels of repeating grids. The new representation offers great versatility, improving the performance on a wide range of tasks. Our method provides greatly increased model capacity compared to fully implicit models while at the same time having a comparably low computational footprint and the same ease of use. Compared to previous grid-based hybrid representations, the LoE model demonstrates good parameter efficiency and generalization capability.

**Limitations.** The nearest-interpolation variant of our model has undefined derivatives and discontinuities when crossing the grid boundaries. Even though the smooth interpolation variants can be used in these scenarios, compared to SIREN, it does not have smooth, high-order derivatives, limiting its use in applications such as solving differential equations. Similar to previous works, our method required prior knowledge of the underlying signal in order to choose suitable grid scales.

**Broader Impact.** Our model is orthogonal to various other approaches of improving or extending the increasingly popular implicit neural representations (INRs), such as better activation functions [39], better input encoding [49], the use of hypernetworks [14, 44, 45], or combination with other pre- or post-processing CNNs [5, 38]. Our method can enable higher fidelity data (image, video, volume, *etc*.) synthesis, representation, and compression at reduced computational costs compared to prior works. Like prior works, our method can be misused to negative ends, including for *deepfakes*.

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
