# Supplementary Material for
# Implicit Neural Representations with Levels-of-Experts

## A  Comparison of Different Hierarchical Grid Layouts

In the main paper, we have shown that our levels-of-experts framework supports a wide variety of grid layouts (Sec. 3), and arranging the grids in a multi-scale fashion improves the performance (Sec. 4.1 and Sec. 4.3). This section further investigates the performance implications of different hierarchical grid layouts on a 2D toy experiment, where we fit the model to a $512 \times 512$ image.

Here we only consider the case of nearest interpolation (piecewise constant) parameterization of the position-dependent weights. In 2D, consider an input coordinate $\mathbf{p} \in [0, 1]^2$, and a $2 \times 2$ weight tile for the $i$-th layer $\{_{0,0}^i, _{0,1}^i, _{1,0}^i, _{1,1}^i\}$, the position-dependent weight $\psi^i(\mathbf{p})$ can be computed as follows:

$$\psi^i(\mathbf{p}) = {}_{\lfloor x \rfloor \bmod 2, \lfloor y \rfloor \bmod 2}^i, \tag{7}$$

$$\text{where } \begin{bmatrix} x \\ y \end{bmatrix} = \mathbf{A}^i \mathbf{p} + \mathbf{b}^i. \tag{8}$$

An affine transformation $\mathbf{A}^i$ and $\mathbf{b}^i$ is selected for each layer to allow the weight grids to cover a wide range of spatial frequencies.

We study the effect of different arrangements of $\mathbf{A}^i$ and $\mathbf{b}^i$ via controlled experiments. As shown in Table 5, our models outperform the PE MLP baseline over a wide range of hierarchical grid layouts. In this paper, for consistency, we mainly use coarse-to-fine arrangements in all the experiments, similar to the *Quad Tree* arrangement in this case. However, we also want to point out that the performance of the LoE model can be further improved by tuning the grid layouts, as evident by the better performance achieved with the *Fine to Coarse* arrangement in this case.

**Architectures.**  For all the experiments, we use a 10-layer network with 64 hidden channels, which consists of 9 position-dependent linear layers $i = 1, \ldots, 9$ and a final linear layer. All of the position-dependent linear layers use a $2 \times 2$ weight tile. We use Leaky ReLU (0.2 negative slope) activation function between all the layers. We use positional encoding [10] with $L = 8$ frequencies as the input mapping.

**Training Details.**  We use the mean squared error as the reconstruction loss. We use Adam optimizer with $(\beta_1, \beta_2) = (0.9, 0.995)$ and a learning rate of 0.005. We train all the models for 10k iterations, with the learning rate decayed to 0.0005 after the first 5k iterations. We initialize the bias vectors to zeros and initialize the weights using the uniform distribution variant of Kaiming initialization [3]. We sample the full image in each iteration (i.e. no subsampling).

**Dataset.**  We use the "camera" test image from the scikit-image Python package [1], which is a grayscale image with a resolution of $512 \times 512$. This image is CC0 licensed.

**Runtime & Hardware.**  Each experiment takes approximately 7 minutes on a NVIDIA Titan V GPU.

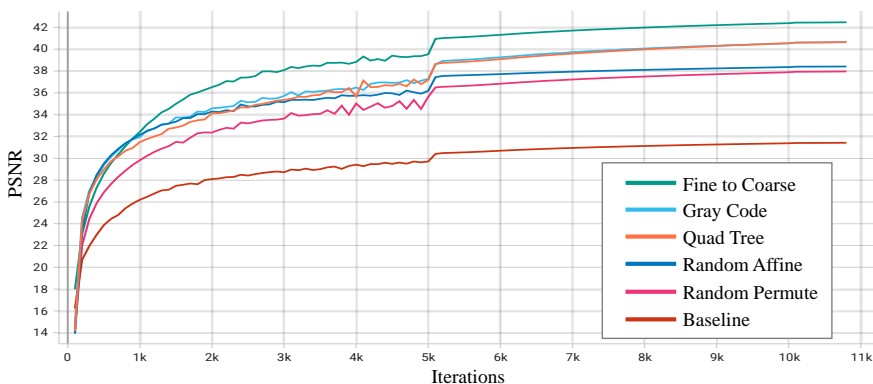

Figure 8: PSNR vs training iterations curve for the 2D toy experiment.

Table 5: Comparison of different hierarchical arrangements of weight grids for a network with 9 position-dependent linear layers and $2 \times 2$ weight tiles. The PE MLP baseline is also included in the first row. $\mathbf{I}$ denotes the $2 \times 2$ identity matrix.

| | Visualization | $\mathbf{A}_i$ | $\mathbf{b}_i$ | PSNR(dB) |
|---|---|---|---|---|
| PE MLP | | $0\mathbf{I}$ | $\mathbf{0}$ | 31.38 |
| Quad Tree | | $2^i\mathbf{I}$ | $\mathbf{0}$ | 40.55 |
| Gray Code | | $\begin{cases} 2\mathbf{I} & i = 1 \\ 2^{i-1}\mathbf{I} & i > 1 \end{cases}$ | $\begin{bmatrix} 0.5 \\ 0.5 \end{bmatrix}$ | 40.55 |
| Fine to Coarse | | $2^{10-i}\mathbf{I}$ | $\mathbf{0}$ | 42.40 |
| Random Permute | | $2^{\mathrm{arr}[i]}\mathbf{I},$ $\mathrm{arr} = [3,8,1,9,6,2,5,4,7]$ | $\mathbf{0}$ | 37.89 |
| Random Affine | | $\begin{bmatrix} a_i & b_i \\ c_i & d_i \end{bmatrix},$ $a_i, b_i, c_i, d_i \sim \mathcal{N}(0, 16^2)$ | $\begin{bmatrix} m_i \\ n_i \end{bmatrix},$ $m_i, n_i \sim \mathcal{U}(0, 1)$ | 38.36 |

# B  Details for the Image Fitting Experiments

**Architectures.**  For all the experiments, we use 8-layer networks with [512, 384, 256, 256, 256, 256, 256, 3] respective output channels. We use Leaky ReLU with a negative slope of 0.2 as the activation function between each layer. We use positional encoding with $L = 13$ frequencies as the input mapping for all the experiments. For the coordinate embedding baseline, we bilinearlly interpolate a learnable 128-channel $256 \times 256$ embedding and feed it to the first layer of the MLP, concatenated with the encoded input coordinates. For experiments using position-dependent weights, we use $4 \times 4$ weight tiles for the first 7 layers and use a regular linear layer for the last layer. The per-layer affine transformation coefficients for these experiments can be found in Table 6. They are defined in the same way as in Sec. A.

Table 6: Per-layer affine transformation coefficients of weight grids for image fitting experiments that use position-dependent weights. All the models have 7 position-dependent linear layers ($i = 1, \ldots, 7$) and use $4 \times 4$ weight tiles. The input 2D coordinates have a range of $\mathbf{p} \in [0, 1]^2$. † This prevents the grid pitch from becoming finer than the pixel pitch.

| | $\mathbf{A}_i$ | $\mathbf{b}_i$ |
|---|---|---|
| Interleaved | $8192\mathbf{I}$ | $\mathbf{0}$ |
| Chunked | $4\mathbf{I}$ | $\mathbf{0}$ |
| Random Affine | $\begin{bmatrix} a_i & b_i \\ c_i & d_i \end{bmatrix},$ $a_i, b_i, c_i, d_i \sim \mathcal{N}(0, 256^2)$ | $\begin{bmatrix} m_i \\ n_i \end{bmatrix},$ $m_i, n_i \sim \mathcal{U}(0, 1)$ |
| Constant Input (no PE) | $\begin{cases} 4\mathbf{I} & i = 1 \\ (4^{i-1} \times 2)\mathbf{I} & i > 1 \end{cases}$ † | $\mathbf{0}$ |
| Ours Bilinear | $\uparrow$ (same as above) | $\mathbf{0}$ |
| Ours | $\uparrow$ | $\mathbf{0}$ |

**Training Details.**  We use Adam optimizer with $(\beta_1, \beta_2) = (0.9, 0.999)$ and a learning rate of 0.001. For each iteration, we randomly sample 262144 pixels from the image. We train the models for a total of 200k iterations. We decay the learning rate with a multiplier of 0.1 after 100k iterations. We follow the same network initialization scheme described in Sec. A.

**Runtime & Hardware.**  All the experiments are run on a single NVIDIA Tesla V100 GPU (with power consumption capped at 163W). The training time is 8 hours for *PE*, 9 hours for *PE + CE*, 39 hours for *Ours Bilinear*, and 10 hours for the rest of the models.

**Dataset.**  In this experiment, we use the public domain image of the dwarf planet Pluto[1] (NASA/Johns Hopkins University Applied Physics Laboratory/Southwest Research Institute/Alex Parker). The original image is $8000 \times 8000$ pixels. We resize it to $8192 \times 8192$ following the published implementation of [9].

**Uncertainties of the Quantitative Results.**  We report uncertainties of the quantitative results in Table 7, which extends Table 1 in the main paper, with standard deviations over multiple runs included.

# C  Details for the Video Fitting Experiments

**Architectures.**  For all the experiments, we use 6-layer networks with 256 hidden channels, with the exception of the SIREN-L experiment, which has 1024 hidden channels. We use Leaky ReLU

---

[1] https://solarsystem.nasa.gov/resources/933/true-colors-of-pluto/

Table 7: Quantitative results for the image fitting experiment. Standard deviations over multiple runs are parenthesized.

|  | PSNR dB (STD) ↑ | SSIM (STD) ↑ |
|---|---|---|
| PE MLP [10] | 32.34 (0.02) | 0.869 (6e-4) |
| PE + CE [2] | 39.65 (0.21) | 0.967 (1e-3) |
| Interleaved | 33.80 (0.01) | 0.876 (7e-5) |
| Chunked | 43.13 (0.01) | 0.980 (7e-5) |
| Random Affine | 42.08 (0.78) | 0.973 (5e-3) |
| Constant Input (no PE) | 39.48 (0.06) | 0.955 (4e-4) |
| Ours Bilinear | **44.85 (0.02)** | **0.985 (1e-4)** |
| Ours | **44.24 (0.17)** | **0.983 (4e-4)** |

with a negative slope of 0.2 as the activation function between each layer. We use positional encoding with $L = \log_2 512 = 9$ frequencies as the input mapping for PE, PE + CE and LoE experiments. For the PE + CE baseline, similar to the image fitting experiment, we use a 64-channel $64 \times 64 \times 64$ embedding that is trilinearlly interpolated and fed to the first layer.

For the LoE experiment, we use $2 \times 2 \times 2$ weight tiles for the first layer, $4 \times 4 \times 4$ tiles for the next 4 layers, and use a regular linear layer for the last layer. The grid resolution for the 5 position-dependent layers are 2, 8, 32, 128, and 512, respectively. Their corresponding affine coefficients are $\mathbf{A}_i = $ grid resolution $\times \mathbf{I}$ and $\mathbf{b}_i = \mathbf{0}$. The 3D input coordinates are normalized to $\mathbf{p} \in [0, 1]^3$.

**Training Details.**  We use Adam optimizer with $(\beta_1, \beta_2) = (0.9, 0.999)$. For PE, PE + CE, and LoE experiments, we use a learning rate of $0.001$. For SIREN experiments, we use a learning rate of 5e-5 for stable training. For each iteration, we randomly sample 160000 pixels from the video. We train the models for a total of 200k iterations, decaying the learning rate with a multiplier of 0.1 after 100k iterations. For SIREN experiments, we follow the initialization scheme in [12]. For the rest of the experiments, we follow the same network initialization scheme described in Sec. A.

**Runtime & Hardware.**  All the experiments are run on a single NVIDIA Tesla V100 GPU (with power consumption capped at 163W). The training of PE, PE + CE, and SIREN experiments require 3 hours, while the LoE model requires 4 hours due to the inefficient implementation of the dynamic weight layer. The SIREN-L experiment requires 26 hours of training.

**Dataset.**  The original video is permissively licensed and can be found here[2]. We use the cropped and downsampled version from [12].

**Uncertainties of the Quantitative Results.**  We additionally report the standard deviations of the quantitative results in Table 8, which corresponds to Table 2 in the main paper.

**Additional Results.**  We include the result videos from all the methods in the supplemental material package.

# D   Details for the Novel View Synthesis Experiment

**Architectures.**  We use an identical network architecture skeleton, as shown in Fig. 9, in all the experiments. We use Leaky ReLU with a negative slope of 0.2 as the activation function between each layer. We use a positional encoding with $L = 10$ frequencies as the input mapping for the coordinates ($\gamma(\mathbf{p})$). We follow the standard settings [10] and use a positional encoding with $L = 4$ for the ray directions ($\gamma_d(\mathbf{d})$).

---

[2]https://www.pexels.com/video/the-full-facial-features-of-a-pet-cat-3040808/

Table 8: Quantitative results of the video fitting experiment, with standard deviations reported in the parenthesis.

|  | PSNR dB (STD) |
|---|---|
| PE [10] | 27.33 (0.01) |
| PE + CE [2] | 35.83 (0.20) |
| SIREN [12] | 29.13 (0.004) |
| SIREN-L | 37.71 (0.01) |
| Ours | **39.98 (0.12)** |

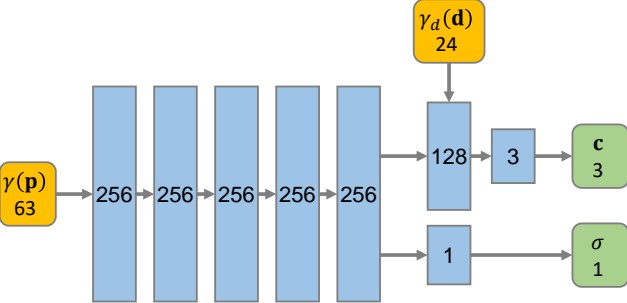

Figure 9: Network architecture for the novel view synthesis experiment. Numbers denote output channels. Yellow blocks denote inputs, while green blocks denote outputs.

For the LoE experiment as well as the *No Hierarchy* ablation, we replace the first 5 layers with position-dependent linear layers and use $4 \times 4 \times 4$ weight tiles in these layers. The grid layouts are shown below in Table 9. For all the experiments, we use identical network architectures for both coarse and fine networks.

Table 9: Affine transformation coefficients for novel view synthesis experiments.

|  | $\mathbf{A}_i$ | $\mathbf{b}_i$ |
|---|---|---|
| No Hier. | $64\mathbf{I}$ | $\mathbf{0}$ |
| Ours | $4^i\mathbf{I}$ | $\mathbf{0}$ |

**Training Details.**  We use Adam optimizer with $(\beta_1, \beta_2) = (0.9, 0.999)$. We use an initial learning rate of 5e-5, and decay it exponentially following the published implementation of [10]. We follow the same network initialization scheme described in Sec. A. For each iteration, we randomly sample 16384 rays. For each ray, we sample 96 coarse samples and 192 important samples. Each model is trained for a total of 500k iterations.

**Runtime & Hardware.**  All the models are trained using $8\times$ NVIDIA A40 GPUs. The LoE model, and the ablation model requires 26 hours of training, while the PE MLP baseline requires 19 hours.

**Dataset.**  We use the preprocessed version of Tanks and Temples dataset [6, 8] for training, which includes 133 training images and 19 test images with a resolution of $1920 \times 1080$. The dataset is CC-NC licensed. We isotropically scale the scene so that everything is bounded within $[-0.5, 0.5]^3$ in the world coordinate.

**Uncertainties of the Quantitative Results.**  We additionally report the standard deviations of the quantitative results in Table 10, which corresponds to Table 3 in the main paper.

**Additional Results.**  We also include result images evaluated from test views in the supplemental material package.

Table 10: Quantitative results for the novel view synthesis experiment. Standard deviations are reported in the parenthesis.

|  | PSNR dB (STD) ↑ | SSIM (STD) ↑ |
|---|---|---|
| PE MLP [10] | 30.50 (0.09) | 0.900 (1e-3) |
| No Hier. | 27.73 (0.66) | 0.861 (0.012) |
| Ours | **31.46 (0.04)** | **0.936 (1e-3)** |

# E   Details for the Image Generation Experiment

**Architectures.**   In the image generation experiment, we adopt a linearly arranged network with residual connections, as shown in Fig. 10. We use a positional encoding with 8 frequencies as the input mapping, and apply Leaky ReLU to the activations. For the *PE + CE* baseline, we use a 256-channel $256 \times 256$ embedding. For the LoE experiment, we use position-dependent weights on the first linear layer of 7 intermediate residual blocks, with the grid resolutions and tile sizes marked in the figure. The definition of these parameters follows Sec. C. The above setting yields approximately the same parameter counts and computational costs, enabling fair comparison. This network architecture is inspired by the "residual" setting used in [2] but greatly simplified and scaled down to accelerate the experiments.

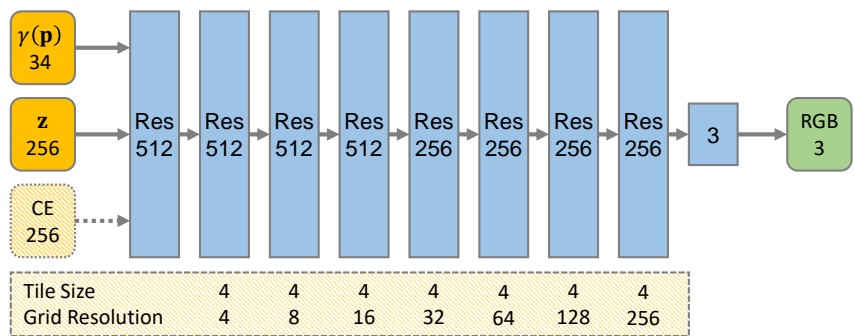

Figure 10: Network architecture for the GAN experiment. Blocks labeled with "Res" are residual blocks. Numbers on the blocks denote output channels. The coordinate embedding (CE) input is used only for the CE experiment, while the position-dependent weight tiling, with tile sizes and grid resolutions noted under each block, is only used in the LoE experiment.

For the discriminator network, we use simple 5-layer patch discriminators [4] constructed with kernel size 3 and stride 2 convolution layers and Leaky ReLU activation functions. We use two such discriminators at two different scales: full image and 1/2 downsampled image. We use spectral normalization [11] on the discriminator weights.

We use hinge loss [7] as the GAN objective. The overall losses for the discriminator and the generator are as follows:

$$L_D = -\mathbb{E}_{\mathbf{x} \sim p_{data}} \left[ \min \left( 0, -1 + D \left( \mathbf{x} \right) \right) \right] - \mathbb{E}_{\mathbf{z} \sim p_{\mathbf{z}}} \left[ \min \left( 0, -1 - D \left( G \left( \mathbf{z} \right) \right) \right) \right] \tag{9}$$

$$L_G = -\mathbb{E}_{\mathbf{z} \sim p_{\mathbf{z}}} D \left( G \left( \mathbf{z} \right) \right) \tag{10}$$

Table 11: Image generation quality on FFHQ dataset, with standard deviations over two runs provided in the parenthesis.

|  | FID (STD) |
|---|---|
| PE + CE [2] | 23.5 (0.7) |
| Ours | **18.3 (1.6)** |

**Training Details.** We use Adam optimizer with $(\beta_1, \beta_2) = (0.9, 0.999)$ and a learning rate of 1e-4. We follow the same network initialization scheme described in Sec. A for the generators and discriminators. We use a batch size of 4 images per GPU, yielding a combined batch size of 32 images per iteration. Each model is trained for 1M iterations.

**Runtime & Hardware.** Both models require approximately 60 hours of training on $8\times$ NVIDIA A100 GPUs.

**Dataset & Evaluation Metrics.** The models are trained on the Flickr Faces-HQ (FFHQ) dataset [5], which contains permissively licensed images from Flickr that were intended for free use and redistribution by their respective authors. In our experiments, we downsample the images from $1024 \times 1024$ to $256 \times 256$. We report FID scores that are evaluated with a sample size of 50k images.

**Uncertainties of the Quantitative Results.** We additionally report the standard deviations of the FID scores in Table 11, which corresponds to Table 4 in the main paper.

**Additional Results.** We include additional examples of generated images in Fig. 11.

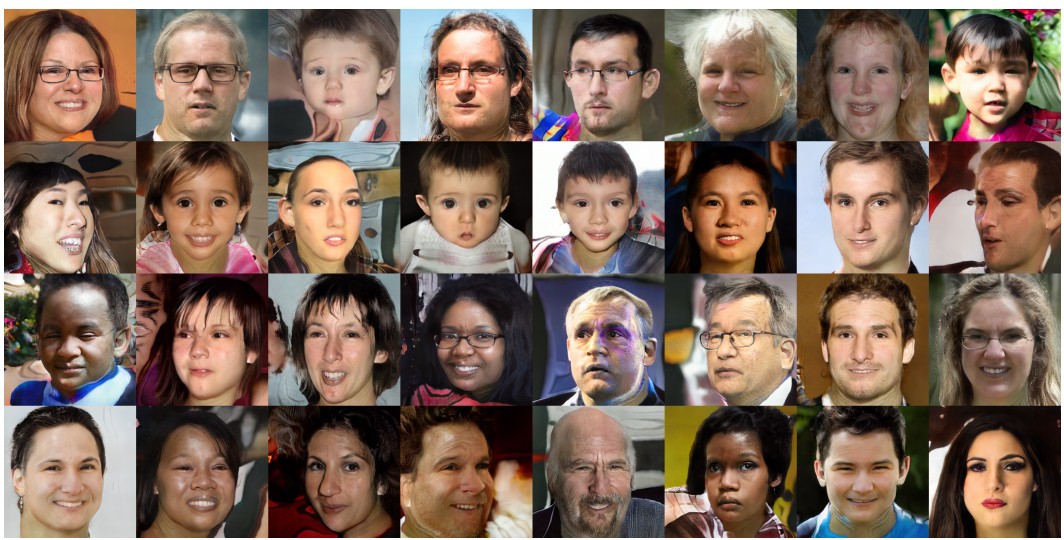

(a) Levels-of-Experts

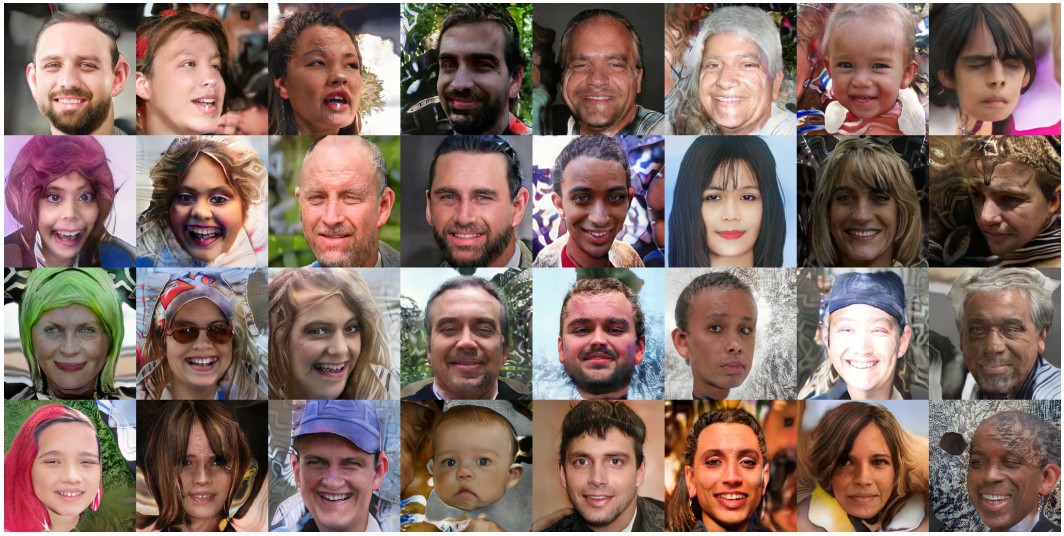

(b) Coordinate Embedding

Figure 11: Additional image generation results from our method (a) and baseline method (b).