# OpenReview forum: "Implicit Neural Representations with Levels-of-Experts"
_NeurIPS.cc/2022/Conference — NeurIPS 2022 Accept_

### Official Review · Reviewer_Girn · 2022-07-05

**Rating:** 5
**Confidence:** 2
**Soundness:** 3 good
**Presentation:** 3 good
**Contribution:** 2 fair

**Summary:**

The paper proposes an algorithm for performing implicit neural representation in a hybrid fashion, trading off between the discrete and continuous counterpart through a hierarchical tiling structure. The authors propose a fast tiling mechanism determining which MLP system to use at what hierarchical level (layer). The authors showcase that this methodology of using multiple experts at every layer with a fast expert-accessing mechanism leads to performance benefits on modelling high resolution images and videos, and ultimately outperforms the baseline counterparts on the tasks of novel view synthesis as well as GAN-based image generation tasks.

**Questions:**

I have a few clarification questions that might help me understand the work better.

* Why is there such a big difference in the number of parameters between PE MLP system and the others?
* Further, why are the number of parameters same in PE + CE with the proposed models? The proposed model introduces N different weight matrices at every layer, so I would assume that the number of parameters would grow by $\times N$
* In the novel view synthesis, how is only *indirect supervision* provided? Is there not ground-truth supervision based on the true color and density at the different points on the ray? Some clarification on this would be helpful.

**Limitations:**

The work can be substantially improved by averaging the performance of the proposed algorithm over multiple different data-points as opposed to just a single example. Further, the authors should do some form of variance analysis to make sure that the results are statistically significant. Finally, for some reason a lot of the Figures have captions that talk about top and bottom row, while the figure only has one row (eg. Figure 4)

Beyond this, there are no potential negative societal impacts that I can think of which stem from this work.

**Strengths And Weaknesses:**

**Strengths**

* The authors propose an intuitive extension to coordinate-based MLP systems by allowing for coordinate-dependent weight matrices.
* The choice of expert chosen at any point in the grid and hierarchy is obtained through an extremely fast algorithm.
* The results show improvements of the proposed approach over the baselines, often with lower memory cost.

**Weaknesses**

* The proposed work introduces certain additional hyper-parameters, namely $(\alpha^i, \beta^i)$, ablations regarding which are lacking in the draft.
* The authors claim that their approach leads to dynamic expert selection; however the proposed algorithm only lays out a static tiling based on which expert selection is done. I would recommend the readers to refer to *Is a Modular Architecture Enough?* (Mittal et. al 2022) or *Recurrent Independent Mechanisms* (Goyal et. al 2019) as some of the works that look into *dynamic* expert selection, as opposed to a static grid tiling.
* Are the results of the experiments in Section 4.1 - 4.3 averaged over different high resolution images or different video samples or is it only based on a single example?
* The work is lacking in quantifying the variance of the results, which can be obtained through multiple seeded runs as well as multiple data-points.

---

> ### Author Response · Authors · 2022-08-01
> **Response to Reviewer Girn**
>
> **Large difference in #parameters between PE MLP and others:** For fair comparison, the numerical performances of the different models are reported using the same backbone network architecture (identical hidden channel counts and network depth, thus similar computational cost). Given the same network architecture, an LoE model or a model with CE can utilize much more parameters compared to a PE MLP, leading to the difference in parameter count.
>
> **Why do PE+CE and LoE models having the same #parameters:** Unlike PE MLP or SIREN, both models are able to decouple parameter count and computational cost. Thus for fair comparison, besides using the same network architecture, we also configure them to have the same parameter count. The parameter count of an LoE model can be adjusted by the use of different tile sizes. For PE+CE model, it can be adjusted by modifying the resolution and channel count of the coordinate embedding grid.
>
> **Indirect supervision in the novel view synthesis experiment:** In this experiment, the coordinate-based network models a 3D radiance field while the ground-truth data contains only 2D images. Instead of directly supervising the 3D color and density, we render the radiance field to 2D and then apply photometric loss there. We are supervising a 3D model with 2D data, hence the name indirect supervision. The use of this terminology is in line with the previous work [23].
>
> **Ablation of $\alpha$ and $\beta$ ($A$ and $b$ in higher dimension):** The experiments in Table 1 can be considered as an ablation of $(A_i, b_i)$, with corresponding parameters listed in Supp. Table 5. Additional ablations are included in Supp. Table 4.
>
> **Static vs dynamic expert selection:** We would like to thank the reviewer for the references. In LoE, although the experts are assigned according to the input coordinate, they are indeed following a fixed rule. It will be an interesting future work to extend the current framework with a learned expert selection scheme. We will revise the terminology in the next revision.
>
> **Experiments in Section 4.1 - 4.3 and their variances:** The results of each task are based on a single high-resolution data, averaged over 3 runs (with standard deviations reported in the supplemental material). We believe that being able to perform well across multiple domains is already a strong enough evidence to showcase the good performance of our model. Besides, an additional set of image experiments is provided in supp. A.
>
> **Caption of Fig. 4:** The "second row" in the caption of Fig. 4 refers to the zoom-in views on the bottom of the figure. We will make them more distinctive in the next revision.

---

> > ### Comment · Reviewer_Girn · 2022-08-07
> > **Additional Information**
> >
> > Thanks to the authors for providing clarifications regarding most of my concerns. I also see that the authors have provided boiler-plate code as a proof-of-concept, but I am curious to know if the authors will be open-sourcing their code, if published?

---

> > > ### Author Response · Authors · 2022-08-08
> > > **Code release**
> > >
> > > Yes! Our intention is to release the code, including the optimized CUDA kernels, on GitHub.

---

> > > > ### Comment · Reviewer_Girn · 2022-08-08
> > > > **Thanks for Response**
> > > >
> > > > Thanks to the authors for all the clarifications. I have updated the score to reflect my updated views on the work. I still believe, though, that a static tiling might be quite a heuristic choice, and some experiments should be done on dynamic selection of experts.

---

### Official Review · Reviewer_9VaW · 2022-07-08

**Rating:** 7
**Confidence:** 2
**Soundness:** 3 good
**Presentation:** 3 good
**Contribution:** 3 good

**Summary:**

The submission proposes a new Neural Architecture for coordinate-based networks.
Motivated by previous work that decomposes the domain and trains distinct networks within them, a so-called"Level of Experts" architecture is proposed.
Instead of distinct networks for each subarea, the authors propose to assign each weight matrix of a layer to multiple parts of the signal domain with periodic recurrence via a query function. The number of available weight matrices and the frequency of their occurrence can vary for each layer of the architecture, i.e. the result is a hierarchy of position-dependent and periodic weights.
This is accomplished via first shifting and scaling the position, and then passing it through a modulo-based operation that maps the position to a grid of weight matrices.
To avoid discontinuities, the query function does not return a single weight matrix, but instead a linear or bilinear interpolation of the 'nearest' (with respect to the query function) weight matrices.
Notably, the proposed architecture does not require an additional positional encoding at the input.

Experiments:
1.  The expressivity of the architecture for fitting a high-resolution (8192x8192) image as well as a video (512x512; 300 frames) is demonstrated. Comparisons are made with related architectures such as MLPs with positional encoding and MLPs with sinusoidal activations (SIREN).
Different settings for the query function are evaluated, each of which corresponds to different "patterns" in the signal space.
2.  A Novel View Synthesis task ("Tanks and Temples dataset") using NERF. Comparisons include MLPs+positional encoding and a nonhierarchical setting (i.e. a decomposition of the domain into chunks for independent networks). Focus is drawn to the spatial artifacts of the simple domain decomposition, whereas the proposed method resolves these issues with the interpolation of the weights.
3. Image Generation using GANs, with the proposed architecture as the generator.

In all settings of 1.) and 2.) the "Level of Experts" architecture provides higher PSNR and SSIM while having comparable computational cost to a simple MLP with positional encoding.
Similar results are obtained for the generative model in 3.), here showing a higher FID with comparable computational costs.
For the interpolation between the weight matrices, the linear interpolation shows similar performance to the bilinear case, while having significantly lower computational costs.

**Questions:**

The method of [1] is mentioned to have a similar high-level idea, and its limitations are indirectly shown with the chunked settings.  I think the argument for the advantage of the "Levels-of-Experts" could be made even stronger, by highlighting that the architecture of [1] requires the "Chunked" network to be distilled from a pre-trained network.

[1] Reiser, Christian, et al. "Kilonerf: Speeding up neural radiance fields with thousands of tiny mlps." Proceedings of the IEEE/CVF International Conference on Computer Vision. 2021.

**Limitations:**

The authors sufficiently address the limitations in terms of the required prior information about the signal for selecting the grid scales.
In addition, the limited applicability to settings that require (higher-order) derivatives (e.g. Physics Informed Neural Networks) is mentioned, which is a result of the underlying interpolation method between the weight matrices.

**Strengths And Weaknesses:**

### Significance:
- (Medium) An general architecture for coordinate-based Neural Networks is proposed, outperforming comparable state-of-the-art in a wide range of tasks while preserving low computational costs.
- (Medium) The proposed framework is mostly orthogonal to existing methods for improving implicit representations and appears to be relatively simple to implement, offering itself to practical use and future work.

### Originality
- To the best of my knowledge, the proposed use of periodically reoccurring weight matrices in each layer of the "Levels-of-Experts" is novel.
- The related work seems to sufficiently cover the existing literature and competitive methods. It should however be noted that I have limited familiarity with this specific subfield and might not be aware of newer work.

### Quality
- The paper seems to the best of my knowledge to be technically sound, however, it is mostly of a practical nature providing a novel architecture/framework.
- The experiments cover a wide range of tasks, and are generally sound, showcasing both quantitative and qualitative results and advantages compared to existing approaches.
- No code is included, although later publication is indicated. This disagrees with the given answer in Checklist 3. a).

### Clarity
- The paper is well written, clearly structured, and in general enjoyable to read.
- The method is clearly described. It should be clear how to reimplement the proposed levels of experts given the paper.

In summary, the paper is well written and the proposed method is sound.
The contribution is of mainly practical nature, but its clear advantages are shown in extensive experiments.

---

> ### Author Response · Authors · 2022-08-01
> **Response to Reviewer 9VaW**
>
> **Highlighting the advantages of LoE over KiloNeRF:** We briefly mentioned the drawbacks of [1] on L107. We will emphasize this in the next revision.
>
> **Continuity of derivatives:** Please refer to the common questions section.
>
> **Code for implementing LoE:** We apologize for the error in the checklist. Here we present a proof-of-concept Pytorch implementation of 1D position-dependent linear layer, which is the building block for a LoE network. Note that in our actual implementation, these operations are fused into a single CUDA kernel for better performance. Some additional implementation details can be found in L68 and L153.
>
> ```
> def positional_dependent_linear_1d(weight, bias, in_feats, in_coords, alpha, beta):
>     r"""Linear layer with position-dependent weight.
>     Assuming the input coordinate is 1D.
>     Args:
>         weight (N * Cout * Cin tensor): Tile of N weight matrices
>         bias (Cout tensor): Bias vector
>         in_feats (B * Cin tensor): Batched input features
>         in_coords (B * 1 tensor): Batched input coordinates
>         alpha (scalar): Scale of input coordinates
>         beta (scalar): Translation of input coordinates
>     Returns:
>         out_feats (B * Cout tensor): Batched output features
>     """
>     B = in_feats.size(0) # Batch size
>     N = weight.size(0) # Tile size
>     Cout = weight.size(1) # Out channel count
>     Cin = weight.size(2) # In channel count
>     # In the actual implementation, the following lines are fused into a CUDA kernel.
>     tile_id = torch.floor(alpha * in_coords + beta).long() % N
>     out_feats = torch.empty([B, Cout])
>     for t in range(N):
>         mask = tile_id == t
>         sel_in_feats = torch.masked_select(in_feats, mask).reshape(-1, Cin)
>         sel_weight = weight[t]
>         sel_out_feats = sel_in_feats @ sel_weight.T
>         out_feats.masked_scatter_(mask, sel_out_feats)
>
>     return out_feats + bias
> ```

---

> > ### Comment · Reviewer_9VaW · 2022-08-08
> > **Response**
> >
> > Thank you for the response and especially for the code snippet.
> >
> > To avoid misunderstandings regarding my "limitations" section: \
> > I'd like to highlight, that the issues regarding higher order derivatives were clearly communicated in the paper - I just reiterated the general limitations of the proposed method.

---

### Official Review · Reviewer_SVJM · 2022-07-10

**Rating:** 7
**Confidence:** 4
**Soundness:** 3 good
**Presentation:** 3 good
**Contribution:** 3 good

**Summary:**

The paper addresses the compactness and efficiency issue in grid based signal representations and propose a Levels-of-Experts (LoE) framework that hierarchically arrange the weights of the MLP from a set of candidate weight matrix for high fidelity data (image, video, volume) representation.

**Questions:**

I do not fully understand why the proposed representation is able to reduce computational costs compared to prior works (L296). If I understand it correctly, the linear interpolation between weights is extremely consumed (k times number of layer times nodes of each layer), and the computational costs can be significantly larger for high dimensional data (dim>1).

**Limitations:**

Limitations have been discussed in detail.

**Strengths And Weaknesses:**

What's good:
1) The paper organization and presentation are clear.
2) The idea of position-dependent weight is novel and interesting.
3) The paper provides qualitative and quantitated evaluations on image and video regression, radiance field reconstruction and image generation. The experiments are sufficient to demonstrate the effectiveness.

To be improved:
1) The abstract claims that the key advance of the proposed method is its compact global latent representation compared to most recent grid based approaches, however, I couldn't find any related discussion or experiment on why global representation is better. Maybe a continue or high order derivative could be one of the advantages?
2) More comparison results and details discussions with recent approaches would be appreciated, such as BACON[23], MFN [9] and instant-NGP [33].
3) Seems the reference to TensoRF is incorrect, please check.

---

> ### Author Response · Authors · 2022-08-01
> **Response to Reviewer SVJM**
>
> **Reduced computational cost compared to prior works:** It is true that the linear interpolation of weight matrices is expensive (Table 1 "Ours Bilinear", L207). However, thanks to the use of hierarchical weight tiling, our LoE model with nearest interpolated weights performs similarly well compared to linearly interpolated models (Fig. 2, Table 1), saving computational cost considerably.
>
> **Why global representation is better:** Having a compact global latent leads to strong performance in generative modeling, as demonstrated in section 4.4. This is not possible with grid-based methods such as Instant-NGP. Moreover, compared to previous grid-based methods, LoE tends to have a better parameter efficiency as it captures repeating patterns more efficiently.
>
> **Continuity of derivatives:** Please refer to the common questions section.
>
> **Comparison with BACON and MFN:** MFN is loosely related to LoE in that it can also be considered as a model using position-dependent weights. However, it shares the same drawback with SIREN and PE MLP that it is expensive to scale up. We implement two versions of MFN using Gabor and Fourier filters and compare them with LoE on fitting the 8192*8192 image used in Section 4.1. All the models have the same network depth and hidden channels.
> For Gabor MFN, it achieves a PSNR of 25.53 and a SSIM of 0.766. The Fourier MFN is slightly worse at 24.88 and 0.7621. Both MFNs are significantly worse than LoE while taking more than double the time to train. Although the fitting quality is worse, we note that MFNs have smooth (higher-order) derivatives, which can be useful for some tasks.
> BACON is a more constrained version of Fourier MFN which enforces band-limiting by freezing the frequencies. This is different from the goal of LoE thus we only include results from MFN.
>
> **Comparison with Instant-NGP:** Please refer to the common questions section for detailed discussions.
>
> **TensoRF reference is broken:** Thanks for the suggestion. We will fix it in the next revision.

---

> > ### Comment · Reviewer_SVJM · 2022-08-09
> > **Further question**
> >
> > I appreciate the effort authors bring to the revision especially given the short period of rebuttals.
> > With the nearest interpolation (the "ours" row in Table 1), does it equal a multi-scale version of kilonerf?

---

> > > ### Author Response · Authors · 2022-08-09
> > > **Connection to KiloNeRF**
> > >
> > > KiloNeRF and the nearest interpolated version of LoE model share a similar idea of partitioning the input space and assigning a different set of network parameters to each partition. However, LoE generalizes KiloNeRF in terms of how the space is partitioned and how the network parameters are derived.
> > >  * In KiloNeRF, there is only a single resolution grid. The partitioning is done on a network level. In LoE, each layer of the network has its own grid at a different resolution, promoting smoothness.
> > >  * In KiloNeRF, each grid cell has an independent set of network parameters and they are not repeated. In LoE, the parameters for each layers are tiled and repeated, improving parameter efficiency and enabling generative modeling.
> > >  * We demonstrate that LoE has great flexibility in terms of the shape of the grids. Good performance can be achieved even with random affine grids.

---

### Official Review · Reviewer_yDin · 2022-07-12

**Rating:** 8
**Confidence:** 4
**Soundness:** 4 excellent
**Presentation:** 4 excellent
**Contribution:** 3 good

**Summary:**

A Mixture of Experts framework is proposed that uses a dynamic weight MLP with position-dependent weights for solving various coordinate based tasks such as signal fitting, novel view synthesis and generative modeling. Unlike other hybrid representations that divide the space to use different MLPs, the proposed method uses layer-level tiling in the network architecture which produces smoother boundary interpolations. The results adequately establish the efficacy of the approach on 4 challenging tasks compared to strong baselines.

**Questions:**

1. How did you come up with the optimal hierarchical tiling pattern.
2. What are the ranges of alpha and beta. How do these hyperparameters affect the continuity at the boundaries.

**Limitations:**

The limitations have been adequately addressed in the paper.

**Strengths And Weaknesses:**

Strengths:
1. The paper is very well-written, with relevant references and easy-to-follow narration.
2. The method is simple and general enough such that can be plugged into many different coordinate based tasks, as shown in the experiments.
3. The evaluations show significant gains in accuracy as compared to the prior art.

Weaknesses:
1. The paper claims that the LoE formulation could be used to model large-scale signals. None of the experiments reflect this claim.
2. The network design of hierarchical and tiled weights seem handcrafted. It is not clear if there is a way to come up with the correct pattern without trial-and-error.
3. It will be interesting to see comparison to more recent encoding strategies such as Instant-NGP's multi-resolution hash encoding etc.

---

> ### Author Response · Authors · 2022-08-01
> **Response to Reviewer yDin**
>
> **Optimal tiling pattern:** We arrange the tiles in a power-of-two growing fashion (L167, Fig. 1(a)), which empirically performed well across all the tasks and outperformed non-hierarchical tiling (Fig. 2). However, we also note that this is not always optimal. For example, in the study shown in Supp. Table 4,  the inverted "fine to coarse" arrangement actually performed better than the default "quad tree" arrangement in that specific task, suggesting that there is room for improvement. We hope that this can be explored in the future work.
>
> **Ranges of $\alpha$ and $\beta$:** There is a wide range of selections for $\alpha$ and $\beta$ (or matrix $A$ and vector $b$ in higher dimensions). They determine the tiling patterns of the layers. In supp. Table 4, we provide a non-exhaustive list of possible $A$ and $b$ arrangements and their corresponding performances. In practice, $A$ and $b$ can be arbitrary matrices and vectors, even randomly sampled ones.
>
> **How do $\alpha$ and $\beta$ affect continuity:** Using a wide range of $A$s and $b$s in different layers can promote continuity. This can be seen in Fig. 2, where hierarchical tiling (a) gives more uniform error distribution than (c), which uses identical $A$ and $b$ for all layers. Similar effect can be observed in Fig. 5.
>
> **Modeling of large-scale signals:** We consider "large-scale signals'' to be signals of high resolution. For example, our the experiment of fitting a 8192*8192 pixel image is already considered large-scale in previous works such as ACORN [25].
>
> **Comparison with Instant-NGP:** Instant-NGP lacked a compact latent code and cannot be used directly in generative tasks. For fitting a single signal, Instant-NGP is faster but less parameter efficient than LoE. Please refer to the common questions section for detail.

---

> > ### Comment · Reviewer_yDin · 2022-08-08
> > **Thanks for the update**
> >
> > Other than the question on the tiling pattern (which I and other reviewers raised), I think the paper is a solid contribution, and I maintain my initial score of strong accept.

---

### Author Response · Authors · 2022-08-01
**Response to All Reviewers on Common Questions**

We would like to thank all the reviewers for their valuable comments. Here we address questions common to multiple reviewers. Other reviewer-specific responses are provided under each review section.

**Continuity of derivatives (SVJM, 9VaW):** Similar to many hybrid implicit representations, our model lacked continuous (higher-order) derivatives due to the use of nearest interpolation. We will emphasize this limitation in the next revision. It will be an interesting future direction to bring continuous derivatives to a LoE model without increasing the computation.

**Comparison with Instant-NGP (yDin, SVJM):** Unlike LoE, Instant-NGP lacked a compact latent code and thus cannot be used directly in generative tasks, such as the one shown in Sec. 4.4.
As for representing a single signal, we compare an Instant-NGP of identical parameter count to our LoE model. The Instant-NGP model is trained on the 8192*8192 image used in Section 4.1. We use a hash table size of 775k entries in order to match the parameter counts in Table 1. Like the LoE experiment, we train Instant-NGP for 200k iterations. The Instant-NGP model achieves a PSNR of 40.33 and a SSIM of 0.970, which falls short compared to the LoE model having a PSNR of 44.24 and a SSIM of 0.983. However, we note that Instant-NGP is ~5x faster in terms of wall clock time. The conclusion is that for signal fitting tasks, Instant-NGP is much faster but not as parameter efficient compared to LoE.
We would also like to note that LoE is orthogonal to Instant-NGP and they can potentially be combined together to form a more powerful model. We leave this to future work.

---

### Meta-Review · Area_Chair_s4RP · 2022-08-27

**Recommendation:** Accept
**Confidence:** Certain

**Metareview:**

This paper presents a framework for position-dependent MLPs where the weights in each layer depend on the input coordinate periodically, with hierarchically tiled periodic weight patterns across successive layers. The paper shows that such models outperform prior work on a variety of tasks involving data representation. The reviewers all agree that the proposed approach is novel and highly effective, and the paper is clear and compelling. I accordingly recommend acceptance.

**Award:**

No

---

### Decision · Program_Chairs · 2022-09-14

Accept